# Oral Cavity and *Candida albicans*: Colonisation to the Development of Infection

**DOI:** 10.3390/pathogens11030335

**Published:** 2022-03-10

**Authors:** Mrudula Patel

**Affiliations:** Department of Clinical Microbiology and Infectious Diseases, School of Pathology, Faculty of Health Sciences, University of the Witwatersrand, Johannesburg 2050, South Africa; mrudula.patel@wits.ac.za; Tel.: +27-(11)-4898518 or +27-(82)-9298960

**Keywords:** candida, oral cavity, oral flora, *Candida albicans*

## Abstract

*Candida* colonisation of the oral cavity increases in immunocompromised individuals which leads to the development of oral candidiasis. In addition, host factors such as xerostomia, smoking, oral prostheses, dental caries, diabetes and cancer treatment accelerate the disease process. *Candida albicans* is the primary causative agent of this infection, owing to its ability to form biofilm and hyphae and to produce hydrolytic enzymes and candialysin. Although mucosal immunity is activated, from the time hyphae-associated toxin is formed by the colonising *C. albicans* cells, an increased number and virulence of this pathogenic organism collectively leads to infection. Prevention of the development of infection can be achieved by addressing the host physiological factors and habits. For maintenance of oral health, conventional oral hygiene products containing antimicrobial compounds, essential oils and phytochemicals can be considered, these products can maintain the low number of *Candida* in the oral cavity and reduce their virulence. Vulnerable patients should be educated in order to increase compliance.

## 1. Introduction

The human oral cavity is a unique site, which becomes colonised with bacteria, fungi, mycoplasma, viruses and even protozoa. The acquisition of these microflora occurs throughout life starting within 6 h of birth [1]. However, the distinct character of the oral cavity dictates the type of microflora able to persist. The specialised surfaces such as papillated tongue, and keratinised and non-keratinised squamous epithelium affect the intraoral colonisation and distribution of microorganisms. Although the physiological factors such as pH, temperature, nutrient, redox potential and gingival crevicular fluid contribute to the establishment of this resident microbiota, saliva is the major contributor. For the most part the host–oral microbiota relationship is harmonious; however, it can alter and disease can occur. The disruption of this stability can occur due to intake of antibiotics, frequent intake of fermentable carbohydrates, trauma or oral surgery, and changes in the hosts defences [1].

While bacteria form a major part of the oral microbiota, fungi, which form a small part of the oral microbiota cannot be ignored. Among these fungi, *Candida* species are the most frequent colonisers in the oral cavity and have adapted to reside as a commensal. Distribution of *Candida* is even throughout the mouth, with the most common site of isolation being the dorsum of the tongue. Bacterial colonisation of the oral cavity and the equilibrium are due to the benign behaviour of many components of human microbiota. On the other hand, *Candida* commensalism is a result of the host’s potent innate and adaptive immune responses that restrict the growth of pathogens on the epithelia [1].

The rate of colonisation increases with immunocompromised status, illnesses such as cancers and intra-oral devices including dentures and orthodontic appliances. The oral cavity can be the source of yeast colonisation of the gut and, through saliva, colonizing yeast can be transmitted to other parts of the body. The carriage rate increases during middle and later life. It has been reported that 45% of neonates [2], 45–65% of healthy children [3], 30–45% of healthy adults [4,5] and up to 74% of older people [5,6,7] carry *Candida* in their oral cavities. If the opportunity is given, *Candida* species can cause oral candidiasis, which usually occurs in immunocompromised individuals and those with predisposing conditions. Therefore, *Candida* infections have been named the “disease of the diseased” [1]. In the oral cavity, overgrowth of *Candida* can cause discomfort and pain, an altered sense of taste, dysphagia if disseminated to the oesophagus, difficulty in eating and swallowing and consequently poor nutrition. In immunocompromised patients, infection can disseminate through the bloodstream or upper gastrointestinal tract, leading to severe infection with increased morbidity and mortality. The mortality rate from systemic candidiasis is up to 79% [8]. Therefore, for prevention and treatment, it is important to remove or alleviate the predisposing conditions.

## 2. Role of Saliva and Mucosa

Saliva plays a major role in the preservation of oral health. It lubricates the oral cavity and through constant flow and swallowing, mechanically removes excess bacteria and food debris. Therefore, as a first defence, the quantity of saliva is important. Individuals with xerostomia generally have altered microbial community. Torres et al. (2002) found high *Candida* counts and multiple *Candida* species in subjects with xerostomia [9]. Patients with head and neck cancers going through radiation therapy often have hyposalivation and imbalance in oral flora increasing the rate of *Candida* colonisation and infection [10,11,12]. A low salivary flow rate causes intense oral cavity colonisation by *C. albicans* and other yeasts [13,14].

Mucin present in saliva forms part of the protective coating along the oral mucosal surface, thereby preventing adherence of commensals and pathogens, including *Candida.* In addition, negatively charged epithelial cells cause repulsion of *Candida* cells, preventing adherence. Nevertheless, these adverse effects on adherence are normally oppressed by other stronger adherence mechanisms responsible for the colonisation and development of infection [15].

Saliva also contains complex host molecules, which are both harmful and beneficial to the microflora. Proteins and glycoproteins provide primary nutrients to the oral flora. Once established, the consortia of microflora produce enzymes that can break down salivary molecules and free up required nutrients, carbohydrates and amino acids [16]. For example, *C. albicans* can grow in human saliva without the addition of glucose, and in stationary phase, it can survive for more than 400 h [17]. Saliva contains several ions including sodium, potassium, cadmium, chloride, bicarbonate and phosphate. These ions are usually responsible for the buffering properties of saliva. In addition, it contains mucin, proteins and glycoproteins which provide nutrients to the microorganisms, facilitate their adhesion to the oral surfaces, cause aggregation of microorganisms for clearing from the oral cavity and, to some extent, inhibit the growth of some exogenous microorganisms.

Certain components present in saliva apply selective pressure on the resident microbiota controlling growth and survival. Mucin, fibronectin, proline-rich-protein and secretary IgA actually cause agglutination, binding microorganisms, whereas statherin, histatins, α and β defensins, lactoperoxidase, lactoferrin and lysozymes have antimicrobial activity. Among these salivary antimicrobials, only histatins and defensins proved to have antifungal activity [18]. Statherin is known to reverse the morphological state of *C. albicans* from hyphae, a pathogenic state to a blastospore state, which can alter the course of infection [19]. Histatin has well-known in vitro antifungal activity [20,21] and its protective property in keeping the counts of *Candida* low in the oral cavity has been documented [22,23]. For example, reduced salivary flow and anticandidal activity in HIV-infected individuals contributed to them having an increased incidence of oral *Candida* infections [24]. Defensins are a group of broad-spectrum antimicrobial peptides, produced at mucosal level as components of the innate immunity. Defensins are able to recognize the fungal cell wall and disrupt it through membrane permeabilization. They are cysteine-rich peptides of two families, the α-defensins and β-defensins, which are produced by neutrophils and epithelial cells, respectively [25]. However, the complete mechanism of action of defensins is unknown [26]. Of four β-defensins, hBD-2 and hBD-3 are strongly induced in response to infection, particularly by *C. albicans* hyphae [27]. Defensins also act as chemoattractants for dendritic cells, neutrophils and T cells [28]. LL-37 cathelicidin is an antimicrobial peptide produced by the epithelial cells of the oral cavity and it is known to interact with the *C. albicans* cell wall components such as chitin, glucan and mannan, causing inhibition of *C. albicans* adherence to epithelial cells [29]. 

Intact epithelial cells, although considered as a physical barrier preventing infections, also play an active role in the initial immune response to pathogens. They are responsible for the release of inflammatory mediators, including inflammatory cells and proinflammatory cytokines, e.g., IL-1α/β, IL-6 and TNF-α as well as antimicrobial factors. In addition to the release of cytokines, they produce four classes of small antimicrobial peptides, such as small anionic peptides, which require zinc as a cofactor, defensins, linear cationic peptides rich in proline or tryptophan and small α-helical cationic peptides, which lack cysteines [30]. This protective effect of IL-1β in invasive *C. albicans* infection has been demonstrated [31]. *Candida albicans* pathogenicity and the host response at the mucosal surface are summarised in Figure 1 and Figure 2, respectively [32].

## 3. Host Factors and Habits

Local and systemic host factors (listed in Table 1) affect colonisation and the development of oral *Candida* infections [33]. 

In the oral cavity, an increase in *Candida* counts can occur during compromised host immunity, extremes of age, steroid use, cancers and HIV/AIDS. In addition, when normal microbiota are disturbed due to various host factors, including prolonged use of antibiotics, overgrowth of *Candida* can occur. Local microtrauma, poor oral hygiene, radiotherapy for head and neck cancers, cytotoxic chemotherapy, iron deficiency, malnutrition, oral prostheses and the presence of dental caries can increase the risk of colonisation and the development of infection [34]. Oral *Candida* have been shown to increase in ages above 80, untreated carious or prosthetic teeth, salivary low pH and decreased red blood cell count [35].

Smoking is associated with a variety of changes in the oral cavity. Cigarette smoke has effects on saliva, oral commensal bacteria and fungi, mainly *Candida*, which causes oral candidiasis, the most common opportunistic fungal infection in man [36]. How cigarette smoke affects oral *Candida* is still controversial. This brief overview is an attempt to address the clinical findings on the relationship between smoking and oral candidiasis and possible mechanism of pathogenicity. 

### 3.1. Smoking

Cigarette smoke affects saliva, and oral microbiota, including *Candida*. However, the mechanism of the effect of cigarette smoke on oral *Candida* is controversial [36]. Darwazeh et al. (2010) found that tobacco smoking did not appear to increase oral colonisation with *Candida* species in healthy subjects [37], whereas Muzurović et al. (2013) showed that smoking has an influence on oral colonisation with *Candida* species [38]. Similarly, tobacco users were found to harbour elevated levels of *C. albicans* [39]. Smokers are seven times more likely to have oral *Candida*. In addition, smokers with active carious lesions are also more likely to carry oral *Candida* [40]. Water-pipe smoking is no different to cigarette smoking. Therefore, water-pipe smokers and cigarette smokers are at an increased risk of developing oral *Candida* infections [41]. 

Cigarette smoke (CS) is one of the most important risk factors for lifestyle non-communicable illnesses. It is known to affect host immune functions, which predisposes smokers to infections [42,43]. In the oral cavity, normal flora, secretory antibodies and immune cells, such as polymorphonuclear leukocytes, are important in the inhibition of the establishment of *Candida* [44]. *Candida* colonisation is increased in smokers due to the reduced activity of oral polymorphonuclear leukocytes. In addition, smoking reduces gingival crevicular fluid, which carries antibodies and immune cells [45]. The essential components of innate immunity, the NOD-like receptor family pyrin domain, containing 3 (NLRP3) inflammasome and IL-1β, are required for normal immune function. NLRP3-induced IL-1β together with IL-6 have been shown to be critical mediators of antifungal-protective Th17 immuno-inflammatory responses particularly in *Candida* infection [34]. Animal studies have shown that IL-1β-deficient mice have increased *Candida* counts and lower survival rates compared to the normal wild-type mice if exposed to smoke [46]. Similarly, intracellular expression of TNF-α was also reduced significantly after cigarette smoke exposure in the *C. albicans* infection groups [47]. This consequently reduces the host defence against *C. albicans* infection and increases colonisation. In addition, nicotine has a direct effect on *C. albicans* by enhancing the thickness of biofilm and adherence [48].

### 3.2. Dental Caries

*Candida* carriage has been found to be high in individuals with dental caries [49], particularly in children with early childhood caries [50]. Fragkou et al. (2016) found that caries-active children between the ages of 3 to 13 harboured *Candida* more frequently and had a significantly higher number of *Candida* than caries-free children [51]. *Candida* counts of ≥1000 cfu/mL were found in the saliva of Yemeni children with caries [52]. In children with severe early childhood caries, both the child and the mother were highly infected with *Candida* and they were genetically related, suggesting mother to child transmission [53]. 

*Candida albicans* is an acidogenic fungus and heterofermentative, mainly in the presence of high carbohydrate concentrations [54]. Therefore, this microorganism can participate in the tooth demineralization process [55,56]. Although the pathogenesis of *Candida* in the development of dental caries is not established, *Candida albicans* and *Streptococcus mutans*, a cariogenic bacterium together form an increased biomass [57,58]. In addition, *C. albicans* mannans has the ability to bind to glucosyltransferases produced by *S. mutans*, which also facilitate the incorporation of *C. albcans* into the biofilm and promote growth [59,60]. These could be the reason for the increased carriage rate of *C. albicans* in individuals with caries.

### 3.3. Oral Prostheses

Oral prostheses, such as removable partial and full dentures and prostheses placed after corrective surgeries, are known to be risk factors for *Candida* colonisation and hence development of oral infections. In total, 60% to 100% of denture wearers carry *Candida* in their oral cavity [61,62,63,64]. In addition, high numbers of *Candida* counts have been found in the oral cavities of denture wearers compared to non-wearers [65,66]. Dentures decrease the flow of oxygen and saliva to the underlying tissues, creating an acidic, anaerobic environment, which is conducive to *Candida* growth. In addition, surface characteristics: porosity and hydrophobicity of denture acrylic and the denture lining allow adhesion of *Candida* [67,68,69]. 

### 3.4. Cancer Treatment

Chemotherapy and radiotherapy used for the treatment of cancers are cytotoxic. They cause a reduction in saliva flow and mucosal fragility, resulting in mucositis. These patients are susceptible to *Candida* colonisation and oral *Candida* infections [70]. Therefore, they carry high quantities of *C. albicans* and a variety of *Candida* species in their oral cavities [11,71]. This can mainly be due to hyposalivation created by the cancer therapy, particularly in patients with radiotherapy for head and neck cancers [11,72]. In these patients, the change in the oral flora with increase in *Candida,* is noted even after 2 years post-treatment [10]. During cancer treatment, to prevent oral infections, management of the oral cavity is important.

### 3.5. HIV

Immunosuppression in the host is the most important factor in the colonisation of the oral cavity with *Candida*. Oral and pharyngeal candidiasis caused by *C. albicans* is the most frequently experienced infection in patients with HIV/AIDS [73], even in the era of efficacious antiretroviral therapy. High *Candida* carrier rates in HIV patients have been found, compared to those in healthy individuals. For example, in South Africa, the *Candida* carrier rate in HIV-positive patients who were not on antiretroviral therapy (ART), was found to be 81%, compared to the 63% of the HIV-negative group. Fourteen percent of these HIV-positive *Candida* carriers carried more than 10,000 cfu/mL of saliva compared to zero percent of the HIV-negative subjects [74]. Sixty percent of HIV patients still carried *Candida* in their oral cavity even after being on ARVs [75]. In Thailand and India, the carrier rate was reported to be around 65% [76,77]. Carrier rate is influenced by ARV therapy, CD4 counts, other illnesses such as tuberculosis and diabetes mellitus, dental caries, oral hygiene and oral prostheses [75,78,79]. 

Although HIV infection is associated with the dysregulation of a number of immune functions at the mucosal surface, the ability of patients to mount specific antibody secretory responses seems to remain relatively intact until the late stages of infection [80]. The tissue-signalling cytokines IL-17 and IL-22 produced by mucosal Th17 cells, are critical to host defence against oral *C. albicans* infection. They induce expression of antimicrobial peptides and recruitment of polymorphonuclear neutrophils. Mucosal Th17 cells are depleted in HIV-infected patients reducing the protective mechanism against *Candida* infections [81]. Studies in HIV transgenic mice have demonstrated that defective IL-17- and IL-22-dependent induction of innate mucosal immunity to *C. albicans* is important in the susceptibility to oral candidiasis [82]. It has also been shown that in CD4C/HIV MutA transgenic (Tg) mice, dendritic cells (DCs) are normally depleted and have an immature phenotype, consequently low expression of MHC class II and IL-12. Defective CD4+ T cells primarily determined the susceptibility to chronic carriage of *C. albicans* in these Tg mice [83]. 

### 3.6. Diabetes

Patients with diabetes mellitus are also susceptible to opportunistic infections including oral candidiasis due to elevated serum glucose levels and decreased function of the cellular immune system. Salivary hyperglycaemia is one of the main risk factors for *Candida* infection of oral cavity in patients with diabetes, more than 77% of whom suffer from oral candidiasis [84]. Prevalence of *Candida* carriage and the quantity of *Candida* in the oral cavities of diabetic patients (69%) was found to be significantly high, compared to the healthy individuals (48%) [84]. Poor glycaemic control, periodontitis and the use of oral prostheses further increases the occurrence of *Candida* (prevalence and density) in these patients [85,86,87,88,89].

### 3.7. Organ Transplant

Invasive *Candida* infections, particularly bloodstream, esophageal, gastrointestinal and respiratory, are prevalent in an immediate post-transplant period in organ transplant patients [90]. The prevalence during the first 6 months after transplantation can be up to 50% depending on the type of organ transplant [91,92,93]. Up to 80% of esophageal infections in renal transplant patients were found to be due to *Candida* colonisation of the oral cavity and oral candidiasis [91,92]. Although *Candida* carriage may not be significantly high in transplant recipients [94,95], they carry a high number of *Candida* in their oral cavities. Dongari-Bagtzoglou et al. (2009) found that 57% of asymptomatic transplant recipients carried *Candida* in their oral cavities as compared with 50% of age-matched healthy individuals [96]. However, those 57% of carriers had a high number of *Candida.*

## 4. *Candida albicans*

*Candida* species is the most common oral cavity-colonising fungus, which is a unicellular, dimorphic (blastospore and mycelium) eukaryote cell with sexual and asexual reproduction. It contains a cell wall that is external to the cell membrane. The plasma membrane contains large quantities of ergosterol. Many species of *Candida* occur in the oral cavity and are identified during a diseased and commensal state [12,66,71,74,75,97]. The most common species are *Candida albicans*, *C. glabrata, C. tropicalis, C. krusei, C. parapsilosis, C. guilliermondii* and *C. dubliniensis*. Among these *Candida* species, *C. albicans* is the species most frequently isolated from the oral cavity as well as from extraoral sites. In the oral cavity, since non-albicans *Candida* generally coexist with *C. albicans*, their role in the pathogenesis of oral candidiasis has not been established. However, the mixed colonisation with *C. albicans* and *C. glabrata* has been found to enhance fungal invasion and tissue damage [98]. Most *Candida* species are easy to identify using microscopy and culture technique. Individual species can be differentiated using biochemical reactions which are commercially available.

Among many systemic and localised *Candida* infections, oral candidiasis is one that affects both healthy and immunocompromised individuals. It is the most common human fungal infection, especially in early and later life. It manifests as an inflammation of buccal and palatal mucosa, and tongue. The interplay between the immune status, oral tissues, oral environment and microbial factors is responsible for the development of this infection [34]. Oral candidiasis can be classified into five clinical categories: pseudomembranous candidiasis, erythematous candidiasis, chronic atrophic candidiasis, angular cheilitis and chronic hyperplastic candidiasis [34]. 

The common histopathological features are the presence of chronic inflammatory cell infiltration in the tissue just under the infected epithelium and the accumulation of microabscesses around the *Candida* cells [34]. 

### 4.1. Pathogenicity of Candida albicans

The conversion of blastoconidial state to the hyphal state instigated by local and systemic host factors causes tissue invasion and clinical infection. Other virulence factors are: cell surface expression of adhesins, biofilm formation, thigmotrophism, phenotypic switching and hydrolytic enzyme secretion. These pathogenic factors facilitate host recognition, binding to the host cell, other microorganisms and abiotic surfaces, ability to overcome host response, and tissue penetration and degradation. 

Epithelial cells of the mucosa are at the forefront, and first to interact with *C. albicans*. Adherence of *C. albicans* occurs due to the interaction between fungal cell wall components and surface receptors of the host cells. Although structural polysaccharides such as β-glucan, mannan and chitin induce epithelial signalling, target receptors have not been identified. Adhesins responsible for the adherence are mainly hyphal wall protein 1 (Hwp1) and agglutinin-like sequence 1–9 (*ALS1–9*). Multiple host epithelial receptor targets for *ALS* proteins have been identified [99]. *ALS1–5* and *ALS9* are upregulated during mucocutaneous candidiasis [15]. *ALS3p* adhesion is responsible for the adherence of the hyphae form of Candida and *ALS3* is upregulated during oral and vaginal infections [15]. *ALS3* acts as an adhesion, as well as an invasion. Together with heat shock protein Ssa1, it promotes the endocytosis of *C. albicans* into epithelial cells. This occurs through the interaction with the intercellular component E-cadherin and the epidermal growth factor/human epidermal growth factor 2 complex [100,101,102]. Non-viable metabolically inactive *Candida* cells can still be endocytosed. In addition, INT1 adhesin is also responsible for the adherence targeting epithelial integrins. In addition to adherence, INT1 is also involved in hyphae formation [103].

During adhesion of *C. albicans* to epithelial cells, hyphae are induced and hyphae-associated proteins Hwp1 are expressed. Hwp1 is highly expressed during infection in the oral cavity [104]. It acts as a substrate for epithelial transglutaminases, facilitating string covalent links with other epithelial proteins. This allows further adhesion and establishment of *Candida* to the epithelial cells [105]. Active penetration through hyphae is the dominant route of invasion, rather than the endocytosis. Active penetration through physical pressure created by actively growing hyphae and hydrolysis occurs due to the production of Candidalysin (ECE1) and hydrolytic enzymes, such as proteinases (Secretory Aspartic Proteinases—Sap 1 to 10), Phospholipase B1 (Plb1) and the lipase family (Lip1-10). Among 225 proteins secreted by *C. albicans* which facilitate acquisition of nutrient, tissue invasion and damage, Saps are the most extensively studied proteins [106]. Sap1-8p are extracellular, whereas Sap9p and Sap10p remain attached to the fungal cell membrane. However, collectively they degrade many host tissue components and proteins involved in immune defences. Phospholipases increase adherence ability of *C. albicans* to host cells and hence pathogenicity [107]. The hyphae form of *C. albicans* that do not produce phospholipase particularly PLD1, may adhere to the epithelial cells but cannot penetrate the tissues [108].

Through candidalysin, *C. albicans* hyphae cause damage to the epithelial cells and elicit innate immunity [109]. In oral epithelial cells, candidalysin causes cell damage and membrane destabilization through induced calcium ion influx and lactate dehydrogenase release. Hydrolytic enzymes actually cause digestion of epithelial tissues [110,111,112].

Although the adherence and biofilm formation ability of *C. albicans* to biotic and abiotic surfaces contributes towards the acquisition and development of infection, it does not contribute towards the actual pathology. However, biofilm-associated pathogen is protected from disinfectants and antifungal agents, and therefore it has implications in the infection control regime and the treatment.

### 4.2. From Colonszation to Infection

Mucosal epithelial cells distinguish between the harmless blastospore form of *Candida* and invasive hyphae through the activation of mitogen-activated protein kinase (MAPK) immune pathways [113]. These pathway activations occur in two stages. During the commensal state, there is a weak activation of the NF-kB, phosphoinositide 3-kinase (PI3K), c-Jun N-terminal kinase (JNK), extracellular signal-regulated kinase (ERK1/2) and p38 MAPK pathways [114]. This activity is sustained during the commensal state and does not result in initiation of a pro-inflammatory response and epithelial tissue damage. Nevertheless, the invasive hyphal formed together with candidalysin, particularly in large quantities, elicits stronger activation of the same pathways, which subsequently release pro-inflammatory cytokines causing innate immune response including recruitment of neutrophils and macrophages.

Host immunity is the most important factor in the *Candida* colonisation of the oral cavity. Additional factors that contribute towards the level of colonisation are host habits and physiology, trauma, oral prostheses, antimicrobial and chemotherapeutic treatment and reduced salivary antimicrobial peptides. The main factor in the balance commensalism or parasitism of an invading microorganism is the host’s immune response capacity. However, there are few measures that can contribute to an improvement in the immune response, which directs preventative measures to control the number of *Candida* cells and the virulence of the organism. 

In an oral cavity of an immunocompromised individual, the *Candida* counts are generally high and further increase during infection. For example, 42.6% and 14.4% of antiretroviral-treatment-naive HIV-positive patients carried 10^3^ to 10^4^ and >10^4^
*Candida* cells per millilitre of saliva, respectively. These counts were significantly higher than their HIV-negative counterparts [74]. In 58% of denture wearers and 75% of cancer patients with oral prostheses >10^3^
*Candida* per millilitre of saliva were found [65,66]. Up to 70% of cancer patients carry a high number of *Candida* in their oral cavities without an overt infection [71]. 

Many host factors influence the increase in these counts of *Candida* [35]; however, the threshold value that differentiates between colonisation and actual infection is important. Studies have reported controversial threshold values. Epstein et al. (1980) showed that patients with candidiasis had greater than 400 cfu/mL of *Candida* in their saliva whereas carriers had less than 400 cfu/mL [115]. Xu and Hu, (1993) have reported a *Candida* count of 200 cfu as a cut-off point to *Candida* infection [116]. In our experience, healthy individuals have shown to carry up to 10^3^ cfu/mL in their saliva [73] without showing any signs and symptoms. Similarly, symptomatic patients are suggested to have 10^3^ to 10^6^ cfu/mL in saliva [117]. These differences in the results could be due to the method of sample collection, such as oral swabs, whole saliva, stimulated saliva, oral rinses and concentrated rinses. Swabs are operator-dependent and rinses are likely to give higher counts than saliva. Counts in the concentrated rinses can be the highest [118]. With whole saliva samples, which is the most common method, and ROC curve analysis, 270 cfu/mL have shown to be the threshold value [119]. They also showed that in the symptomatic patients the counts can vary depending on the type of oral candidiasis. These threshold values are important, not only during diagnosis of infection but also in the prevention of the development of infection. The immune status of patients cannot be altered, but the *Candida* counts can be kept lower than this threshold by adjusting contributing host factors.

In addition to the counts, virulence of *Candida* is also important particularly in *Candida* cells carried by the vulnerable population. In a case of high virulence, the required infectious dose, or, in oral Candidiasis threshold counts, may be even lower. *Candida albicans* isolated from HIV patients have enhanced adherence ability, and they produce large quantities of secreted aspartyl protease, suggestive of a high virulent state, than isolates from HIV-negative patients [120]. In addition, virulence of *Candida* carried by asymptomatic HIV-positive patients fluctuates and is not related to the CD4 counts [121]. During their carrier state, *Candida* isolated from patients with oral prostheses produced significantly high levels of hydrolytic enzymes and formed hyphae [66]. In addition, their adherence ability was significantly higher than the isolates from healthy individuals. Similarly, *Candida* isolated from cancer patients without any symptoms of Candidiasis produced a higher level of phospholipases compared to the isolates from healthy individuals [71]. A high number of isolates of *Candida* from diabetics produced a high level of proteinase [122]. Besides the high virulence, in these vulnerable individuals, if the counts of *Candida* increase, they can collectively cause increased tissue damage.

It is evident that immunocompromised individuals carry a high number of *Candida* with enhanced virulence. As soon as the balance is disrupted due to the host or environmental factors, infection can occur. Prevention of the development of infection is possible.

## 5. Prevention and Treatment 

Many systemic and topical preparations containing antifungal agents such as fluconazole, amphotericin B, miconazole, nystatin and clotrimazole are available for the treatment of oral candidiasis. Treatment is relatively easy, apart from the side effects, threat of over-use and development of resistance. Considering the fact that in the oral cavity *C. albicans* is carried by a large population of normal healthy individuals and even more by immunocompromised patients, prevention should be the most important approach, rather than treatment. Alternative treatment, such as proven home remedies and phytochemicals, should be considered [123]. In the oral cavity, whatever therapeutic product is used, it is difficult to maintain the therapeutic concentration due to the constant flow of saliva. Therefore, it is important that the subtherapeutic concentrations have some effect on the virulence of the surviving *Candida* cells, providing additional benefit. The primary objective must be to maintain low *Candida* counts and reduce their virulence. Moreover, the additional host factors such as high carbohydrate intake, dental caries and hyposalivation should be identified as they influence the adherence, growth and virulence of *Candida,* and corrective measures should be implemented (Figure 3). This approach is patient-centred and therefore, for compliance, patient education is very important. These lifestyle adjustments can keep the *Candida* counts in check, and some of the measures may maintain a low level of virulence. 

In the oral cavity *C. albicans* can be associated with the biofilm formed by oral bacteria; therefore, the removal of biofilm and oral hygiene are important. Mechanical removal of biofilm from hard and soft tissues, using toothpastes, and the use of antimicrobial mouthrinses can keep the numbers low. Although the chlorhexidine gluconate mouthrinses are considered a gold standard for the oral cavity, they cannot be used on a long-term basis and have side effects. A daily-use mouthrinse containing fluoride and triclosan have shown to reduce *C. albicans* counts by 77% in HIV-positive patients [124]. Mouthrinses containing cetylpyridinium chloride, menthol, eucalyptol and iodine have antifungal activities and can be used to control *Candida* in the oral cavity [125]. They are not very expensive and have no side effects. Patients with HIV, organ transplants, diabetes mellitus and any immunocompromised individuals should be educated and advised regarding the importance of good oral hygiene. In cancer patients, the few weeks before, after, and during radiation and/or chemotherapy are a crucial time, and it is important to keep the *Candida* counts low. Similarly oral and denture hygiene in denture wearers is important. They should be educated regarding oral and denture hygiene, and night-time removal of dentures.

Consumption of refined sugars should be avoided because *Candida*-associated oral *Streptococcus mutans* ferment sugars, resulting in the production of acids and extracellular polysaccharides, which facilitate adherence and growth, and enhance hyphae and hydrolytic enzyme production in *Candida*. In addition, sugars will also support the growth of *Candida* [60]. 

Probiotics, for their efficacy towards oral *Candida* and their virulence, have also been studied [126]. The mode of action of probiotics is inhibition through competition for adhesion, secretary metabolites and stimulation of the immune system of the host. 

Many medicinal plant-derived compounds and essential oils have been studied. In vitro and in vivo studies have established the antifungal efficacy of essential oils and their effect on the virulence properties of *Candida*. Some of these plants are *Cinnamomum zeylanicum*, *Coriandrum sativum* and *Cymbopogon nardus* [127,128,129]. *Dodoneae viscosa var. angustifolia* (DVA), a medicinal plant that is traditionally used for oral thrush, has also been extensively studied. At high concentrations, DVA has shown antifungal activity against *C. albicans* isolated from HIV-positive and HIV-negative patients, and at subinhibitory concentrations, it inhibits the virulence properties, such as adherence to epithelial cells and, hyphae and biofilm formation [130]. In addition, DVA-derived flavone also inhibits biofilm formation and acid production by cariogenic bacteria *S. mutans*. This means that DVA would reduce the *Candida* counts and improve the oral hygiene, which would further decrease the *Candida* counts [131]. Nevertheless, beneficial plant-derived compounds can be further explored for their cytotoxic effects and in vivo efficacy and developed into mouthrinses or oral gels [132,133]. In clinical trials, drug formulas prepared from essential oils derived from *Pelargonium graveole* and *Melaleuca alternifolia* have shown good efficacy in denture wearers and HIV patients [134,135,136,137].

## 6. Conclusions

The oral cavity is a unique site where host and microbes, including *Candida,* have a delicate balance. They live in harmony with each other until the balance is disrupted. The factors that affect this relationship are host immunity, physiology and habits. Disruption leads to the development of oral candidiasis in vulnerable individuals. Antifungal drugs are available to treat this infection; however, recurrent infections in this group require frequent use of these drugs, which can lead to the development of drug resistance in this causative fungi *Candida.* In addition, these drugs have side effects. Since the causative factors are well known, corrective measures can be implemented to prevent the development of this infection. Not much can be done about the immune status of the host. However, the physiology and habits of the host can be addressed. Preventative measures can include household remedies, use of mouthrinses and gels, and probiotics. The aim of the preventative measures should be to maintain the low counts (<200 cfu/mL) in saliva and inhibit the virulence of *Candida,* which can be achieved using well-studied readily available mouthrinses used for oral hygiene, and phytochemicals. Essential oils have been extensively studied for their antifungal and antivirulence activity and are readily available. Due to the constant flow of saliva in the oral cavity, antifungal activity in these products in not sufficient and it is difficult to maintain the therapeutic concentration. Therefore, at subinhibitory concentrations, these products should have some inhibitory effect on the virulence properties, such as adherence ability, hyphae and biofilm formation, and the production of hydrolytic enzymes and toxins. These products can have a long-lasting effect, meaning that at therapeutic concentrations, they will kill the *Candida*. As the concentrations are reduced in the oral cavity, the subtherapeutic concentrations would render the surviving *Candida* cells avirulent. 

## 7. Future Research

Alternative therapies can be explored for their use in the prevention and treatment of *Candida* infections in the oral cavity. Many plant-derived extracts and phytochemicals have been studied and screened for their antifungal activities. Studies could be performed with a clear direction and clinical application of the chemical constituents. Studies for novel drug discovery should be separated from the validation of plants and their possible uses such as topical, systemic or oral cavity/vaginal rinses, ointment, etc. If a chemical is studied, to target the oral cavity, one should bear in mind the fast clearance of the therapeutic chemical. Fast-acting chemicals are ideal. MIC studies are not enough because they are performed over 24 h; therefore, time-kill studies are important. The effect on virulence should be studied for long-lasting effects. Substantivity (mucosal absorption and slow release) of the chemical should be studied. For example, chlorhexidine gluconate has a very good substantivity and therefore it has a long-lasting effect. Finally, the cytotoxicity should be studied over a shorter time exposure, instead of a full 24 h, because the exposure in the oral cavity is topical and brief, with minimum systemic absorption of the therapeutic product.

## Figures and Tables

**Figure 1 pathogens-11-00335-f001:**
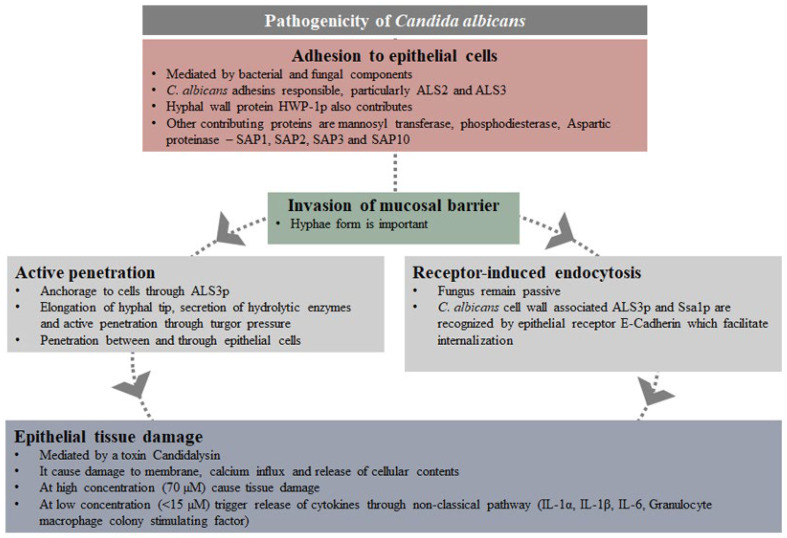
Pathogenicity of *Candida albicans*.

**Figure 2 pathogens-11-00335-f002:**
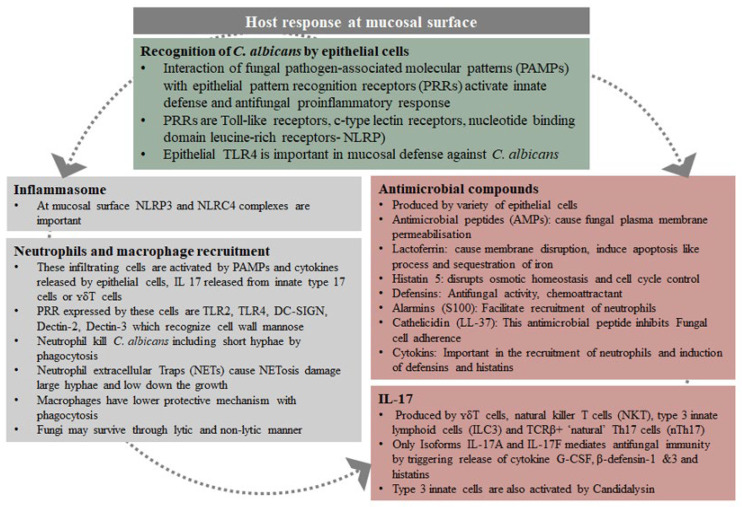
Host response at the mucosal surfaces in *Candida albicans* infections (Summarised from Richardson et al., 2019) [32].

**Figure 3 pathogens-11-00335-f003:**
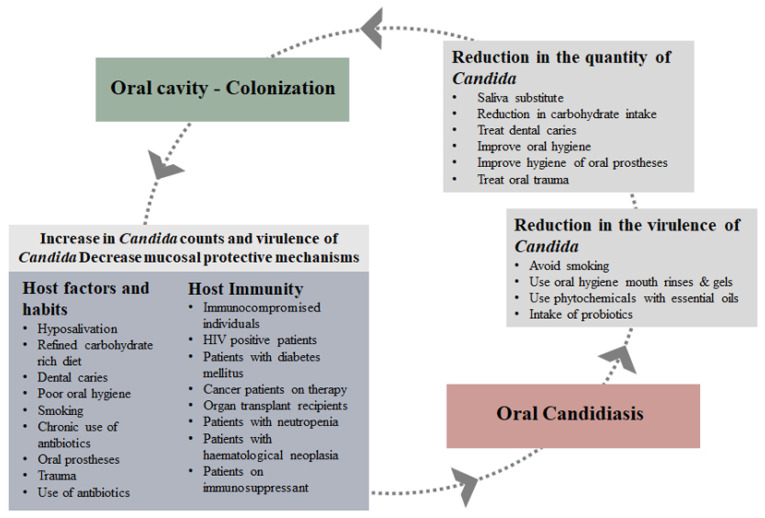
Prevention of development of oral candidiasis.

**Table 1 pathogens-11-00335-t001:** Predisposing factors for oral *Candida* colonisation and infection.

**Local factors and habits**
Hyposalivation Poor oral hygiene Poor denture hygiene and sleeping with dentures Oral prostheses placed due to oral surgery Untreated dental caries Smoking cigarette or waterpipe Microtrauma Infancy and elderly
**Systemic factors**
Radiation treatment for head and neck cancers Malignancies HIV Diabetes mellitus Anaemia and iron deficiency Down syndrome Malnutrition Neutropenia Transplant recipients
**Medications**
Prolong antibiotic use Use of Corticosteroids Cytotoxic chemotherapy for cancers Other immunosuppressive drugs

## Data Availability

Not applicable.

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
