# Peer review of "Oral Cavity and Candida albicans: Colonisation to the Development of Infection"

_pathogens, 2022, doi:10.3390/pathogens11030335_

Round 1
Reviewer 1 Report
The review carried out by Mrudula Patel shows a good compilation of the literature that addresses the subject. However, some points of the manuscript should be better clarified/detailed, which I describe below:
- I suggest that the authors include C. albicans in the title, since throughout the text there are no mentions of other Candida species. The text presented referred exclusively to the species C. albicans.
- Line 29- Please, change “radox” to “redox”
- Line 30: Please, change “residente flora” to “residente microbiota”. In addition, I suggest that you replace on the entire text "flora/microflora" to "microbiota"
- Line 45: “ throught saliva, it can be transmitted to other parts of the body” -Authors need to rewrite this sentence.
- Lines 46- 48: “ The carriage rate increases during middle and later life. It has been reported that 45% of neonates [2], 45-65% of healthy children [3], 30- 45% of healthy adults [4, 5] and 65-88% of those living in care facilities [5, 6, 7] carry Candida in their oral cavities.” The authors refer to the time of life or life stage that have a prevalence of colonization by Candida. However, the last citations refer to the colonization of patients living in care institutions - perhaps the term long-stay institutions would be more appropriate to refer to elderly. To avoid ambiguity, I suggest citing older people/elderly patients and replacing citations with more current references. The ones mentioned are very old.
- Line 86: “Antimicrobial factors present in saliva apply selective pressure on the resident mi- croflora controlling growth and survival”. Please, remove the leading word "antimicrobial" and correct "microflora" to microbiota.
- Line 96: “incidence of oral candida”. Please, change to “incidence of oral Candida..)
- Line 108: “Epithelial cells, although ....”. Please, change to “intact epithelial cells...”
- Line 116: “Candida pathogenicity and the host response at the mucosal surface are summarised in Figure 1 [31]”. The authors could consider including “C. albicans pathogenicity”
- Table 1. Please, write Candida in italics.
- Lines 185-186- format the font
- Line 198: “Chemotherapy and radiotherapy used for the treatment of cancers are intense and cytotoxic”. What is intense? I suggest that the sentence be rewritten.
- Line 201: “Therefore, they carry high quantities of C. albicans and a variety of Candida in their oral cavitie” .Please, include Candida species: Therefore, they carry high quantities of C. albicans and a variety of Candida species in their oral cavitie”
- Line 210: “even in the era of efficacious antiretroviral therapy and consequent immune reconstitution” I suggest the authors revise this sentence. The article by Kelly et al. (Kelly C, Gaskell KM, Richardson M, Klein N, Garner P, MacPherson P (2016) Discordant Immune Response with Antiretroviral Therapy in HIV-1: A Systematic Review of Clinical Outcomes. PLoS ONE 11(6): e0156099), may provide subsidies for the authors in this review.
- Line 257: The most common oral cavity-colonising fungus is Candida species. Please, review the verb agreement of the sentence.
- Line 265- “In the oral cavity, since non-albicans Candida generally coexist with C. albicans, their role in the pathogenesis of oral candidiasis has not been establishe” Not all Candida non-albicans species cohabit with C. albicans in the oral mucosa. Please review the sentence. I suggest reading the work: Lu, S.-Y. Oral Candidosis: Pathophysiology and Best Practice for Diagnosis, Classification, and Successful Management. J. Fungi 2021, 7, 555.
- Lines 342-347: “However, the three fundamental factors that determine the transition from colonisation to infection are the number of Candida cells, virulence of Candida and the host immunity. Host immunity is triggered by the the virulence factors of Candida (hyphae and candidalysin), particularly at the mucosal surfaces. In general, it is difficult to improve immunological functions in patients. Therefore, the focus should be on the number of Candida cells and their virulence” I suggest that the authors re-write the paragraph. Perhaps: the main factor in the balance - commensalism or parasitism of an invading microorganism is the host's immune response capacity. However, there are few measures that can contribute to an improvement in the immune response, which directs preventive measures to control the number of Candida cells and the virulence of the organism.
- Line 356-“ Many host factors influence the increase in these counts of Candida [34] (Nishimaki et al., 2019) “. Please quote appropriately
Author Response
Comments and Suggestions for Authors
Reviewer 1
The review carried out by Mrudula Patel shows a good compilation of the literature that addresses the subject. However, some points of the manuscript should be better clarified/detailed, which I describe below:
- I suggest that the authors include C. albicans in the title, since throughout the text there are no mentions of other Candida species. The text presented referred exclusively to the species C. albicans.
Response: According to the suggestion, Title has been changed.
- Line 29- Please, change “radox” to “redox”
Response: It has been corrected.
- Line 30: Please, change “residente flora” to “residente microbiota”. In addition, I suggest that you replace on the entire text "flora/microflora" to "microbiota"
Response: Flora and microflora has been replaced with microbiota throughout the manuscript.
- Line 45: “ throught saliva, it can be transmitted to other parts of the body” -Authors need to rewrite this sentence.
Response: This sentence has been modified. “The oral cavity can be the source of yeast colonisation of the gut and, through saliva, colonizing yeast can be transmitted to other parts of the body.”
- Lines 46- 48: “ The carriage rate increases during middle and later life. It has been reported that 45% of neonates [2], 45-65% of healthy children [3], 30- 45% of healthy adults [4, 5] and 65-88% of those living in care facilities [5, 6, 7] carry Candida in their oral cavities.” The authors refer to the time of life or life stage that have a prevalence of colonization by Candida. However, the last citations refer to the colonization of patients living in care institutions - perhaps the term long-stay institutions would be more appropriate to refer to elderly. To avoid ambiguity, I suggest citing older people/elderly patients and replacing citations with more current references. The ones mentioned are very old.
Response: It has been modified to elderly people and appropriate references are given. “The carriage rate increases during middle and later life. It has been reported that 45% of neonates [2], 45-65% of healthy children [3], 30-45% of healthy adults [4, 5] and up to 74% of older people [5, 6, 7] carry Candida in their oral cavities.”
- Line 86: “Antimicrobial factors present in saliva apply selective pressure on the resident mi- croflora controlling growth and survival”. Please, remove the leading word "antimicrobial" and correct "microflora" to microbiota.
Response: Corrections have been made. “Certain components present in saliva apply selective pressure on the resident microbiota controlling growth and survival.”
- Line 96: “incidence of oral candida”. Please, change to “incidence of oral Candida..)
Response: Candida is now in italics.
- Line 108: “Epithelial cells, although ....”. Please, change to “intact epithelial cells...”
Response: “Intact” word has been inserted.
- Line 116: “Candida pathogenicity and the host response at the mucosal surface are summarised in Figure 1 [31]”. The authors could consider including “C. albicans pathogenicity”
Response: It has been modified “Candida albicans pathogenicity and the host response at the mucosal surface are summarised in Figure 1 [31].”
- Table 1. Please, write Candida in italics.
Response: It has been rectified.
- Lines 185-186- format the font
Response: Font has been changed.
- Line 198: “Chemotherapy and radiotherapy used for the treatment of cancers are intense and cytotoxic”. What is intense? I suggest that the sentence be rewritten.
Response: “intense” word has been removed.
- Line 201: “Therefore, they carry high quantities of C. albicans and a variety of Candida in their oral cavitie” .Please, include Candida species: Therefore, they carry high quantities of C. albicans and a variety of Candida species in their oral cavitie”
Response: Changes have been made as suggested.
- Line 210: “even in the era of efficacious antiretroviral therapy and consequent immune reconstitution” I suggest the authors revise this sentence. The article by Kelly et al. (Kelly C, Gaskell KM, Richardson M, Klein N, Garner P, MacPherson P (2016) Discordant Immune Response with Antiretroviral Therapy in HIV-1: A Systematic Review of Clinical Outcomes. PLoS ONE 11(6): e0156099), may provide subsidies for the authors in this review.
Response: Suggested publication was reviewed. It was decided that “consequent immune reconstitution” was not appropriate and therefore it was removed. “Oral and pharyngeal candidiasis caused by C. albicans is the most frequently experienced infection in patients with HIV/AIDS [72], even in the era of efficacious antiretroviral therapy.”
- Line 257: The most common oral cavity-colonising fungus is Candida species. Please, review the verb agreement of the sentence.
Response: Sentence has been modified. “Candida species is the most common oral cavity-colonising fungus, which is a unicellular, dimorphic (blastospore and mycelium) eukaryote cell with sexual and asexual reproduction.”
- Line 265- “In the oral cavity, since non-albicans Candida generally coexist with C. albicans, their role in the pathogenesis of oral candidiasis has not been establishe” Not all Candida non-albicans species cohabit with C. albicans in the oral mucosa. Please review the sentence. I suggest reading the work: Lu, S.-Y. Oral Candidosis: Pathophysiology and Best Practice for Diagnosis, Classification, and Successful Management. J. Fungi 2021, 7, 555.
Response: Suggested reference was reviewed however another reference is given. Sentence has been modified accordingly. “In the oral cavity, since non-albicans Candida generally coexist with C. albicans, their role in the pathogenesis of oral candidiasis has not been established. However, the mixed colonization with C. albicans and C. glabrata has been found to enhance fungal invasion and tissue damage [97].”
- Lines 342-347: “However, the three fundamental factors that determine the transition from colonisation to infection are the number of Candida cells, virulence of Candida and the host immunity. Host immunity is triggered by the the virulence factors of Candida (hyphae and candidalysin), particularly at the mucosal surfaces. In general, it is difficult to improve immunological functions in patients. Therefore, the focus should be on the number of Candida cells and their virulence” I suggest that the authors re-write the paragraph. Perhaps: the main factor in the balance - commensalism or parasitism of an invading microorganism is the host's immune response capacity. However, there are few measures that can contribute to an improvement in the immune response, which directs preventive measures to control the number of Candida cells and the virulence of the organism.
Response: Paragraph is rewritten as suggested.
- Line 356-“ Many host factors influence the increase in these counts of Candida [34] (Nishimaki et al., 2019) “. Please quote appropriately
Response: Number has been kept and the name removed.
Reviewer 2 Report
The article is focus in a very interesting issue, mainly for the frequency of this kind of infections in oral cavity. Author made a complete review , well organized and very well explained.
Figures help a lot to understand the problem and bibliography is complete
Author Response
No suggestion given.
Reviewer 3 Report
Manuscript is well written and interesting. Candida presents one of the main pathogen in oral cavity and can cause infection in immunocompromised patients and in patients with dental prosthesis. This review focus in prevention of the development of candidiasis and points out importance of host physiological factors and habits and how good oral health and hygiene can prevent candidiasis. Also, there is an increase in candida resistance to conventional therapy so it is important to share knowledge that alternative therapy like essential oils and phytochemicals can be considered as potential therapeutic agents in candida infection treatment and prevention.
Author Response
No suggestion given.
Reviewer 4 Report
The article is very relevant and presents in great detail a significant problem that affects many groups of people which is oral thrush. I believe that the insertion of a topic addressing the form and methodologies of fungal diagnosis would be an addition that would complete the proposed idea. At times the reading seemed repetitive, focused only on the determining factors for the establishment of colonization . Current treatments used were brilliantly addressed, but new prototypes and treatment candidates deserved a unique topic. The conclusion presented needs to be more objective, as it seems to be starting a new topic to be discussed and not concluding all the discussion carried out in the article.
The figures presented present flowcharts that help to understand the information contained in the text, but the sharpness and excess of text make it confusing for the non-specialist reader. I suggest reformulating and adjusting the sharpness of these images.

Author Response
Reviewer 4
The article is very relevant and presents in great detail a significant problem that affects many groups of people which is oral thrush. I believe that the insertion of a topic addressing the form and methodologies of fungal diagnosis would be an addition that would complete the proposed idea. At times the reading seemed repetitive, focused only on the determining factors for the establishment of colonization . Current treatments used were brilliantly addressed, but new prototypes and treatment candidates deserved a unique topic. The conclusion presented needs to be more objective, as it seems to be starting a new topic to be discussed and not concluding all the discussion carried out in the article.
The figures presented present flowcharts that help to understand the information contained in the text, but the sharpness and excess of text make it confusing for the non-specialist reader. I suggest reformulating and adjusting the sharpness of these images.
Response: Fungal diagnosis: I am not sure it implies clinical diagnosis or a laboratory. A sentence has been added in the section Candida albicans “Most Candida species are easy to identify using microscopy and culture technique. Individual species can be differentiated using biochemical reactions which are commercially available.”
New prototypes and treatment candidates deserved a unique topic: This has not been addressed because it is not clear if the reviewer wants a section inserted or it is just a remark suggesting a topic for future.
Conclusion and Future research has been separated to address the focused conclusion.
Images has been modified according to the Journal requirements.